# Automatic Calibration Diagnosis: Interpreting Probability Integral Transform (PIT) Histograms

## Abstract

Uncertainty quantification in predictive models is essential for safe decision-making and risk assessment. The predictive uncertainty is often represented by a predictive distribution because it is its most general representation. Optimising the sharpness of the distribution subject to its calibration is necessary. This work addresses the proper calibration of predictive distributions in regression tasks. We particularly focus on machine learning models, which are increasingly prevalent in real-world applications. We employ the probability integral transform (PIT) histogram to evaluate calibration quality. It can be used to diagnose calibration problems, e.g. under- or over-estimation, under- or over-dispersion, or an incorrect number of modes. However, PIT histograms are often difficult to interpret because multiple calibration problems may occur simultaneously. To tackle this issue, we present a methodological concept for the automatic interpretation of PIT histograms. It is based on a mixture density network interpreter trained with a synthetic data set of PIT histograms. Given a predictive model, data set, and corresponding PIT histogram, the interpreter can identify a probable observation-generating distribution. This allows us to diagnose a potential calibration problem by comparing the predictive with the probable observation-generating distribution. To showcase the power of the proposed concept in the automatic interpretation of PIT histograms, we referred to regression tasks on standard data sets. As a result, we could achieve notable improvements in the calibration of machine learning models.

## 1 Introduction

Predictive (especially machine learning) models are increasingly prevalent in real-world applications. While they bring many benefits, they are often not perfect and can make incorrect predictions. To enable safe decision-making, risk assessment, and much more, we have to represent (i.e. quantify) and consider the uncertainty of their predictions. Among various means of representing *predictive uncertainties*, probability distributions are their most general representation. However, there is a critical question: How well do these predictive distributions represent those predictive uncertainties?

The key to answering this question is the paradigm of maximising the *sharpness* of predictive distributions subject to their *calibration* (Gneiting et al., 2007). Calibration refers to the statistical consistency between predictive distributions and observations, while sharpness refers to the concentration of predictive distributions. Here, we focus on regression tasks and use the probability integral transform (PIT) histogram as a tool for *calibration diagnosis*. In the machine learning literature, the calibration plot is also a common tool to diagnose calibration (Kuleshov et al., 2018). These two tools are equivalent because both display an estimate of the PIT distribution: the PIT histogram shows a density estimate, whereas the calibration plot displays an estimate of the cumulative distribution function (see appendix A). One should be able to diagnose a potential *calibration problem* by visually inspecting a PIT histogram or calibration plot. However, in some cases, the calibration problem can only be diagnosed if one has a strong familiarity with the behaviour of these tools.

After an introduction to calibration, sharpness, PIT histograms and proper scoring rules (see section 2), we present a methodological concept for an *automatic calibration diagnosis* that is novel

to the best of our knowledge. We train a neural network interpreter with a synthetic data set of PIT histograms to automatically interpret PIT histograms by decomposing them into predictive and probable observation-generating distributions (see section 3). Then, users can see a potential calibration problem by visualising the distributions. Looking at distributions is much more user-friendly than looking at PIT histograms. To showcase the power of the proposed concept, we automatically diagnose the calibration of standard probabilistic machine learning models on standard regression data sets (see section 4).

## 2 Calibration, sharpness, PIT histograms & proper scoring rules

Arguably, the main purpose of a predictive distribution is to quantify the residual predictive uncertainty in a task that cannot be explained when employing a corresponding point prediction. Instead of simply asking for a best guess, we request a full specification of all possible observations and their corresponding probability. Naturally, this specification in the form of a predictive distribution should be reliable, e.g. whenever the model predicts an event to be observed with a probability of 10 %, then we expect the observed event frequency to be 10 % as well. A predictive distribution that satisfies this property is called *calibrated*.

A different predictive model, possibly with access to a richer data set, may issue a different probability for the same observation but still be calibrated. This can happen if the model can better distinguish between events and adapt to each event accordingly. The probability density in the predictive distribution will be more concentrated, and the degree of concentration is called *sharpness*. Both under- and over-confident predictive distributions are undesirable, suggesting that the available information determines the optimal level. Therefore, we aim to maximise sharpness subject to calibration.

Formally, a predictive distribution represented by a predictive cumulative distribution function $F$ is calibrated if the PIT $Z = F(Y)$ follows a uniform distribution, where $Y$ denotes the random observation. The PIT is translation-invariant (shifting predictive distribution and observation in the same direction by the same amount yields the same PIT) and scale-invariant (PIT remains the same when increasing the scale for both the predictive and observation-generating distributions). Whenever the PIT value is close to 0, the observation falls into the left tail of the predictive distribution, whereas a value close to 1 means that the observation falls into the right tail. We diagnose (mis)calibration, by visualising the PIT values $\{z_i = F_i(y_i)\}_{i=1}^n$ from the collection of prediction-observation pairs $(F_i, y_i)_{i=1}^n$ via a histogram (i.e. a PIT histogram).

Simple calibration problems can be identified easily (see Figure 1): a *biased* model has a PIT histogram with a single peak at an edge, an *under-dispersed* model has a bell-shaped PIT histogram, and an *over-dispersed* model has a U-shaped one. However, in the case of multi-modal observation-generating distributions or when more calibration problems co-occur, potential shapes of PIT histograms cannot be enumerated easily anymore, which makes the interpretation of PIT histograms (i.e. calibration diagnosis) difficult or even inaccessible for inexperienced users. Therefore, we have to provide a user-friendly interpretation of PIT histograms from which users can recognise the calibration problem. Subsequently, the users can modify their models (e.g. output a mixture of normal distributions instead of a single normal distribution) and get more reliable results.

The PIT histogram is a useful diagnostic tool but unsuitable when comparing two predictive models. To compare predictive distributions, we employ *proper scoring rules*. In a nutshell, a scoring rule is a loss function for predictive distributions, as opposed to point predictions. A scoring rule is proper if it has the essential property that a predictive distribution that matches the true observation-generating distribution minimises the expected score or loss. Implicitly, that property also means that a proper scoring rule will measure calibration and sharpness jointly. The two most commonly used proper scoring rules are the *negative log-likelihood*, also known as *logarithmic score*,

$$\text{NLL}(f_i, y_i) = -\log f_i(y_i), \tag{1}$$

where $f_i$ denotes the probability density function corresponding to $F_i$, and the *continuous ranked probability score*,

$$\text{CRPS}(F_i, y_i) = \int \left(F_i(x) - \mathbf{1}_{x \geq y_i}\right) \, dx.$$

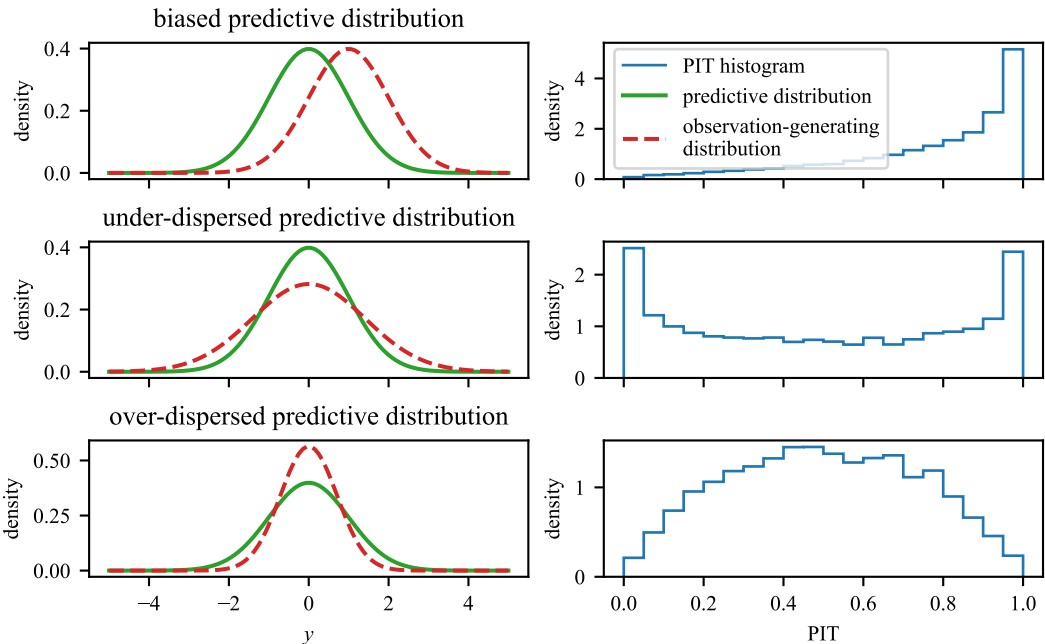

Figure 1: PIT histograms for simple calibration problems. The first row displays a PIT histogram of a *biased* model with a single peak at 1. The second row displays a U-shaped PIT histogram specific for *under-dispersed* models. The last row displays a bell-shaped PIT histogram that refers to *over-dispersed* models. To construct the PIT histograms, we sample observations from the observation-generating distribution and compute PIT values using the predictive cumulative distribution function.

We need to use a proper scoring rule to measure and confirm the improvement in predictive performance that can be achieved by correcting calibration problems. The relevant summary measure is the *mean score*,

$$\frac{1}{n}\sum_{i=1}^{n} S(F_i, y_i),$$

where $S$ is a proper scoring rule. For a more comprehensive overview and review of probabilistic modelling in general and calibration, sharpness, and scoring rules in particular, we refer to Gneiting & Katzfuss (2014, sections 2.2, 2.3, 3.1).

## 3    AUTOMATICALLY INTERPRETING PIT HISTOGRAMS

By automatically interpreting a PIT histogram, we mean its decomposition into an observation-generating and predictive distribution that would lead to the same PIT histogram. This means that we want to represent the inverse function of PIT, including the identification of an observation-generating distribution. We represent the inverse function with a machine learning model called an *interpreter*. The interpreter takes the advantage of PIT histograms that they are invariant to the translation and scale of observation-generating and predictive distribution pairs (see section 2). Therefore, an interpreter trained on a *synthetic data set of PIT histograms* can interpret a given PIT histogram independently of the original translation and scale of the observation-generating and predictive distribution pair. The interpretation allows us to diagnose a potential calibration problem of a predictive model on a data set given its PIT histogram. In the following subsections, we describe the synthetic data set of PIT histograms and interpreter in detail.

### 3.1    SYNTHETIC DATA SET OF PIT HISTOGRAMS

The synthetic data set has to be relevant to the particular application. That means relevant to 1. expected families of observation-generating distributions and 2. predictive distributions that our

models output (e.g. normal distribution). Therefore, the synthetic data set is defined by expected observation-generating distributions, the number of observations $m$ we sample from them, a predictive distribution $F$, and the number of PIT histograms bins $b$. We generate a PIT histogram by

**step 1** choosing an observation-generating distribution, i.e. a specific random variable $Y_i$;

**step 2** sampling $m$ observations $\{y_{i,j}|j = 1,\ldots,m\}$ from $Y_i$;

**step 3** applying PIT to observations, i.e. $F(y_{i,j})$, where $F$ is the predictive cumulative distribution function;

**step 4** binning the PIT values into $b$ bins, i.e. producing a PIT histogram;

**step 5** normalising the PIT histogram (i.e. all bins sum to 1) so that it is independent of the number of observations $m$.

In practice, we would have a single observation $y_i$ sampled from $Y_i$ and single predictive distribution $F_i$ for every input features. However, we do not have input features here; for simplicity, we sample $m$ observations $y_{i,j}$ from every $Y_i$ and have a single predictive distribution $F$ for all observations. Hence, the difference in notation from section 2. We can also use a random number of observations $m$ to generate PIT histograms with various noises. The synthetic data set comprises triplets of an observation-generating distribution, observations, and a PIT histogram.

## 3.2 INTERPRETER

The interpreter is a mixture density network (Bishop, 1994), a machine learning model that outputs a mixture distribution. The input of the interpreter is a PIT histogram. Its output approximates an observation-generating distribution that probably led to the PIT histogram given a predictive distribution. In particular, the interpreter outputs a mixture of normal distributions because it can approximate any observation-generating distribution if it has enough components. It is trained with the synthetic training set using the negative log-likelihood as its loss function. The loss function is computed between a predicted observation-generating distribution (a mixture of normal distributions) and observations from a true observation-generating distribution. This allows observation-generating distributions of the synthetic data set from any family. We illustrate the training in Figure 2.

## 4 EXPERIMENTS

Our experiments showcase that the methodological concept (abstractly described in section 3) can automatically diagnose calibration problems like under- and over-estimation, under- and over-dispersion, and an incorrect number of modes. Currently, the most pressing issue is that uni-modal predictive distributions are used to model multi-modal observation-generating distributions. Therefore, in the following experiments, we focus on under-specified machine learning models that output only a single normal distribution while the observation-generating distribution is multi-modal. First, we evaluate the proposed concept on a synthetic data set. Then, we apply it to real-world regression data sets, where we can automatically diagnose that machine learning models need to output mixture distributions.

## 4.1 EXPERIMENTAL SYNTHETIC DATA SET OF PIT HISTOGRAMS

We choose a simple synthetic data set based on the normal family that allows for the calibration problems mentioned above. Every observation-generating distribution is a mixture of two normal distributions. That means every observation $y_{i,j}$ is a realisation of a random variable $Y_i$, which takes a random value from $N(-d_i/2, t_i)$ with probability $w_i$ or $N(d_i/2, v_i)$ with probability $1 - w_i$. We have the separation $d_i$, weight $w_i$, and variances $t_i$ and $v_i$ parameters. By manipulating them, we can obtain PIT histograms of models that under- and over-estimate, are under- and over-dispersed, or have an incorrect number of modes. We choose to sample $m = 10^4$ observations from the observation-generating distribution. We fix the predictive distribution to standard normal distribution $N(0, 1)$ since the PIT is invariant to translation and scale, and most probabilistic machine learning models output a normal distribution, e.g. Nix & Weigend (1994) or (Lakshminarayanan et al., 2017). Our PIT histogram has $b = 20$ bins.

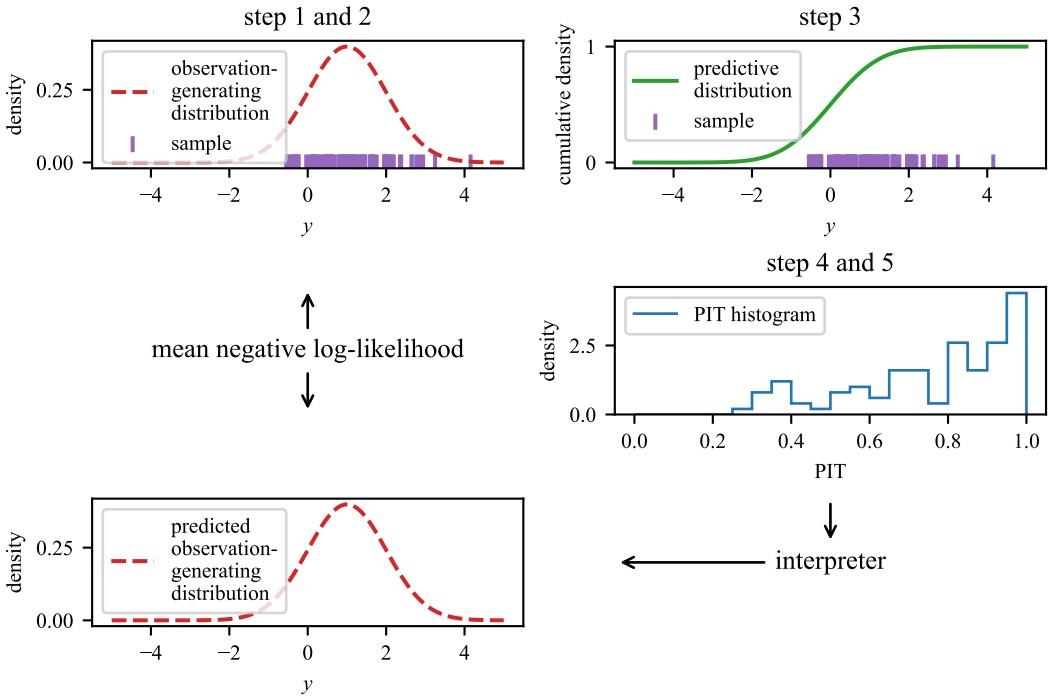

Figure 2: The training of the interpreter starts with sampling an observation-generating distribution. PIT values are computed from them using a predictive distribution. Based on their PIT histogram, the interpreter outputs a probable observation-generating distribution.

If we sample the parameters of the observation-generating distribution, we get an infinite data set. By visually checking the shapes of generated PIT histograms, we decided to sample the parameters from the following distributions. The weight $w_i$ is sampled from uniform distributions $U(0, 1)$. The separation parameter is mainly of interest when generating observations. We use it as a parameter for the severity of ignoring bi-modality. To ensure the sufficient presence of bi-modal scenarios, we sample $u_{i,1}$ from $U(0.1, 1)$ and $d_i = 2(1 - u_{i,1}^2)$. For variances $t_i$ or $v_i$, we sample $u_{i,2}$ and $u_{i,3}$ from $U(-2, 2)$, and $t_i = 2^{u_{i,2}}$ and $v_i = 2^{u_{i,3}}$.

## 4.2 EXPERIMENTAL INTERPRETER

Our experimental interpreter outputs a mixture of 5 normal distributions, which gives the interpreter enough flexibility with respect to our experimental synthetic data set. It has to have 20 input neurons (the PIT histograms have 20 bins), its single hidden layer has 16 neurons, and its output has 15 neurons (weight, mean, and variance for every component). The activation functions are the same as in the original paper by Bishop (1994): the hyperbolic tangent for neurons in the hidden layer, softmax function for weights, and exponential function for variances. We trained it with Adam optimiser in its default setting (Kingma & Ba, 2015), batch size of 100 pairs, and early stopping as described in Algorithm 7.1 in Goodfellow et al. (2016) with patience of 100 epochs, where an epoch is a single batch. As a validation set, we used a fixed random synthetic data set of 1000 triplets generated according to the setting described in section 4.1.

## 4.3 EVALUATION ON EXPERIMENTAL SYNTHETIC DATA SET

First, we test if our interpreter can recover observation-generating distributions at all. We evaluate it on a test set generated as our synthetic data set. The test set contains 1000 triplets, which is enough for statistical significance. For a test PIT histogram, our interpreter predicts an observation-generating distribution. We measure how well it fits the observations that produced the PIT his-

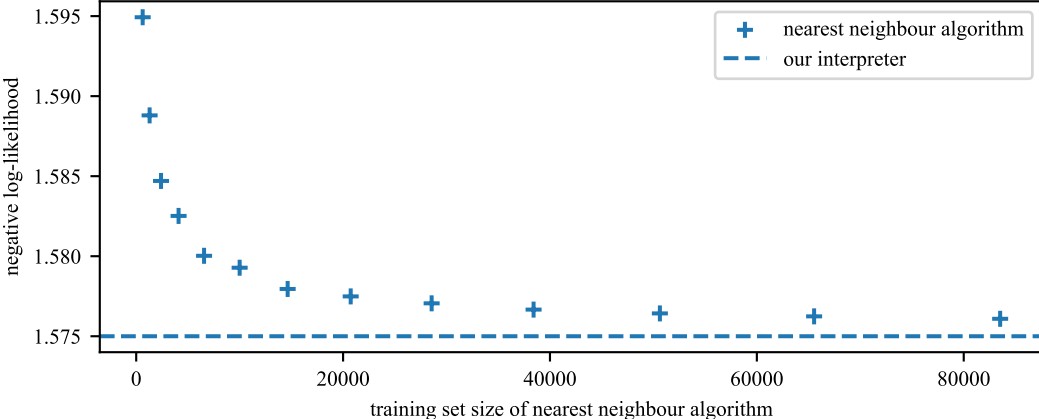

Figure 3: Comparison of our interpreter with the nearest neighbour algorithm for different training set sizes of the algorithm. We only test training sets with sizes less than $10^5$ because larger sets are impractical to generate and store. The plot compares mean negative log-likelihoods on our synthetic test set. Our interpreter can better generalise to new PIT histograms than the nearest neighbour algorithm.

togram with the negative log-likelihood. We report the mean negative log-likelihood on the test set.

We compare our interpreter with the nearest neighbour algorithm, an obvious choice as a simple baseline. We can generate its training set by substituting the uniform distributions of our synthetic data set with values evenly spaced from their minimum to maximum values. For a test PIT histogram, we compute its Euclidean distance from all training PIT histograms. The predicted observation-generating distribution is the observation-generating distribution that produced the training PIT histogram with the closest distance. Moreover, we can control the training set size by the number of evenly spaced values. The number is always the same for all 3 parameters because the parameters are equally important. For example, if we choose the number to be 8, we get a training set with $8^3 = 4096$ triples.

Figure 3 compares the performance of our interpreter trained as described in section 4.2 with the performance of the nearest neighbour algorithm and its dependence on its training set size. Our interpreter can recover its observation-generating distribution with the mean negative log-likelihood of 1.575. It is better than the nearest neighbour algorithm for all tested training set sizes. We do not test training sets with more than $10^5$ triplets because such sets are impractical to generate and store. Our interpreter better generalises to new PIT histograms than the nearest neighbour algorithm.

### 4.4 EVALUATION ON REAL-WORLD DATA SETS

We show that the proposed concept can automatically diagnose the calibration of machine learning models applied to real-world data sets. We choose data sets from the UC Irvine Machine Learning Repository because they are used to evaluate predictive uncertainties, e.g. by Hernandez-Lobato & Adams (2015) or Lakshminarayanan et al. (2017). For our purposes, we limit ourselves to the Year Prediction MSD (Bertin-Mahieux et al., 2011), Physicochemical Properties of Protein Tertiary Structure, and Combined Cycle Power Plant (Tüfekci, 2014) data sets (hereafter *year*, *protein*, and *power* data sets respectively). We will see that they are sufficient to show the power of our methodological concept.

On each data set, we train a *density network* (a simple baseline model with a normal predictive distribution), *deep ensemble* (an advanced model also with a normal predictive distribution) introduced by Lakshminarayanan et al. (2017), and *mixture density network* (a simple model with a more complex predictive distribution). We implement the density network as a mixture density network with a single component. It has a single hidden layer with 100 neurons for the protein and year data sets and 50 for the power data set. Again, activation functions are the same as in the original paper (see

Table 1: Comparison of models in terms of the mean negative log-likelihood (mean NLL) and the mean continuous ranked probability score (mean CRPS). As expected from the automatic calibration diagnoses, mixture density networks perform significantly better for the year and protein data set, while the gap is not that significant for the power data set.

| data set | model | mean NLL | mean CRPS |
|---|---|---|---|
| year | density network | $3.373 \pm 0.003$ | $4.322 \pm 0.013$ |
| | deep ensemble | $3.367 \pm 0.003$ | $4.294 \pm 0.014$ |
| | mixture density network | $3.094 \pm 0.002$ | $4.040 \pm 0.007$ |
| protein | density network | $2.805 \pm 0.039$ | $2.342 \pm 0.025$ |
| | deep ensemble | $2.675 \pm 0.023$ | $2.196 \pm 0.028$ |
| | mixture density network | $2.086 \pm 0.017$ | $1.940 \pm 0.019$ |
| power | density network | $2.795 \pm 0.018$ | $2.175 \pm 0.030$ |
| | deep ensemble | $2.809 \pm 0.017$ | $2.125 \pm 0.032$ |
| | mixture density network | $2.673 \pm 0.023$ | $2.093 \pm 0.042$ |

section 4.2). The deep ensemble comprises the density networks and has 5 ensemble members, as recommended in Algorithm 1 in Lakshminarayanan et al. (2017). The mixture density network has 5 components analogous to deep ensembles. We leave 10 % of a data set for a test set. The rest is split into a training (90 %) and validation (10 %) set. The models are trained with the setting of our interpreter. However, models process the whole training set in every epoch, not a single batch. Features and observations are standardised to zero mean and unit variance. This experimental setting is inspired by Hernandez-Lobato & Adams (2015) and (Lakshminarayanan et al., 2017). However, we only match it partially because we aim to diagnose calibration problems rather than compete with them. Nevertheless, our results in Table 1 are almost identical to those of Lakshminarayanan et al. (2017).

Figure 4 shows the diagnoses on the year data set. The PIT histograms of the density network and deep ensemble are not uniform, so the models are not calibrated. However, it is not clear what the calibration problems are from the PIT histograms. Our interpreter suggests that the calibration problems are combinations of over-estimation and over-dispersion. It is probably caused by the normal predictive distribution that is insufficiently flexible in its shape, i.e. we need a more flexible predictive distribution that can explain the skewness. This interpretation is supported by the fact that the PIT histogram produced by the probable observation-generating distribution is almost the same as the true PIT histogram. The probable observation-generating distribution would be better modelled with a more complex predictive distribution. Here, we use a mixture of normal distribution for simplicity. Indeed, the PIT histogram of the mixture density network is almost uniform, so it is almost calibrated.[1]

The diagnosis on the protein data set is similar to the year data set (see Figure 5). The only difference is that the models under-estimate.

The situation is quite different for the power data set. Looking at Figure 6, we observe that PIT histograms of the density network and deep ensemble exhibit some noise but remain largely uniform. It is plausible that the underlying observation-generating distribution deviates only slightly from a normal distribution. Consequently, the PIT histogram of a mixture density network retains a similar appearance. This suggests we may not anticipate a significant improvement with the mixture density network.

To support the conclusion of the previous diagnoses, we computed the mean NLL and CRPS for our models. We expect the mixture density network to perform better on the year and protein data set than the other models. While on the power data set, we expect all models to perform almost the same. We split the data sets into 5 train-test folds to get standard errors of the means. We show the metrics in Table 1. The metrics of the mixture density network are always better than the metrics of other models. However, as expected, the gap is significant for the year and protein data set, while the gap is not that significant for the power data set.

---

[1]We cannot diagnose the mixture density networks because our interpreter presupposes single normal distributions as predictive distributions, while the mixture density networks output mixtures of normal distributions.

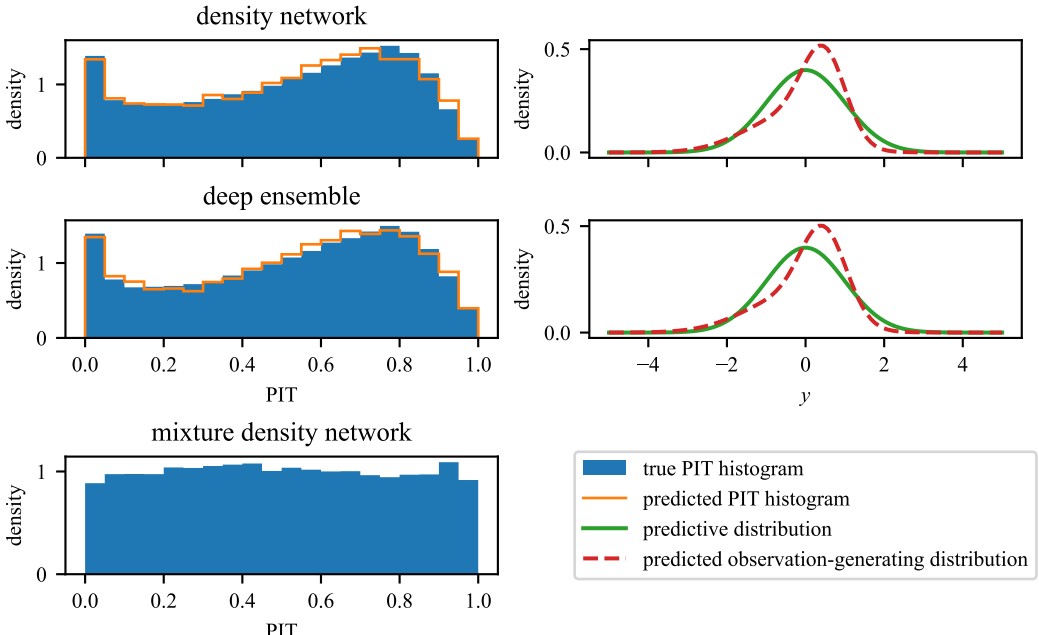

Figure 4: Automatic calibration diagnoses on the *year* data set reveal calibration problems of the density network and deep ensemble, evident from their non-uniform PIT histograms. It suggests inadequate predictive distributions cause their problems. A mixture density network exhibits an improvement in calibration because it has a more complex predictive distribution, as seen in its nearly uniform PIT histogram.

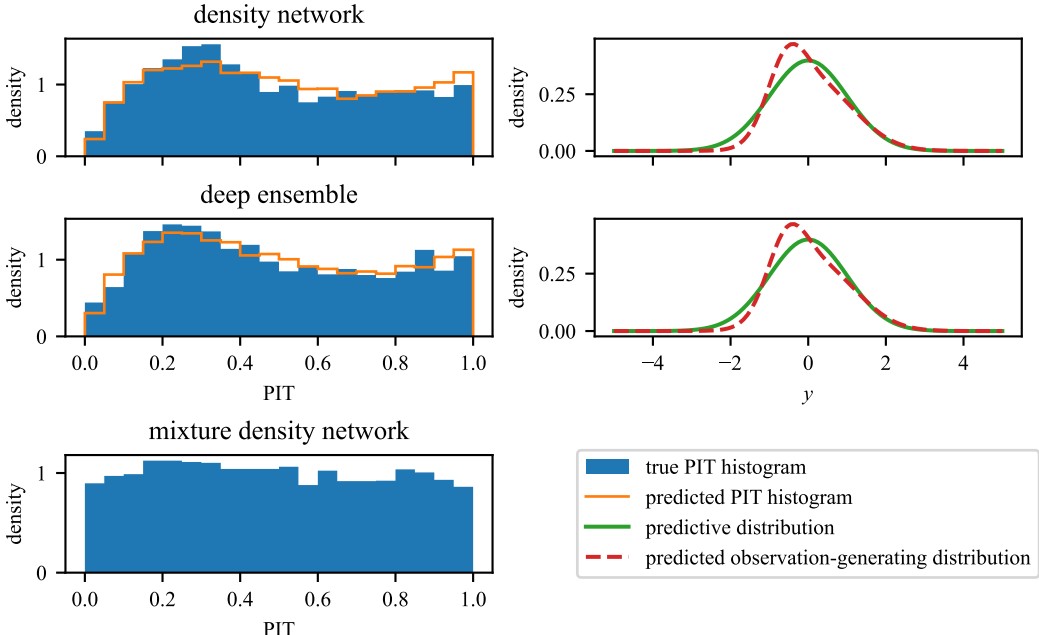

Figure 5: Automatic calibration diagnoses on the *protein* data set lead to the same conclusions as on the year data set (see Figure 4).

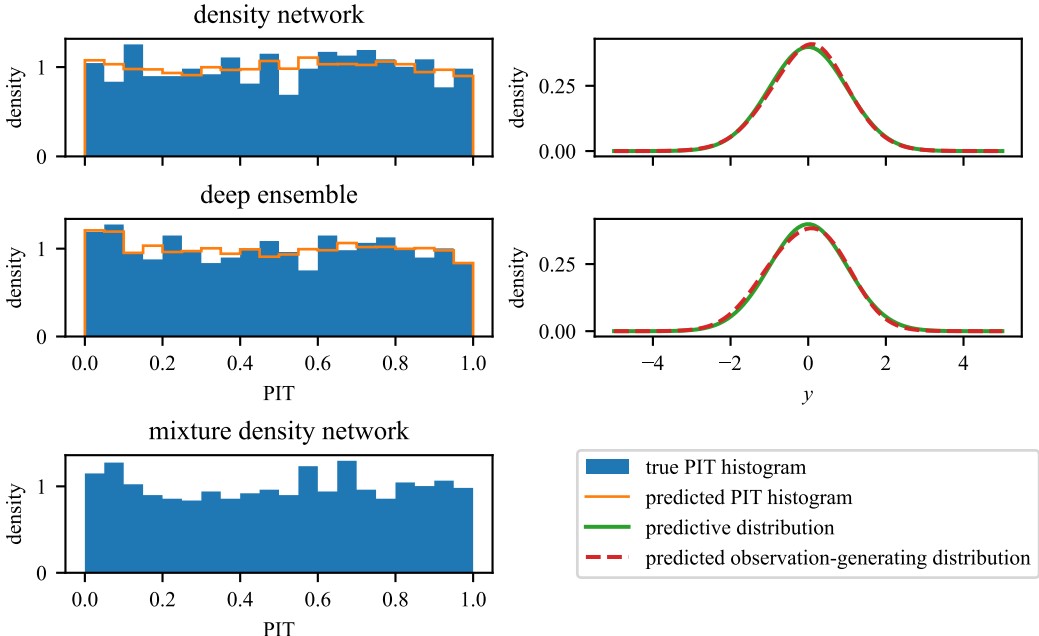

Figure 6: Automatic calibration diagnoses on the *power* data set reveal that all models are almost calibrated.

We showed that the proposed concept can automatically diagnose calibration problems on real-world data sets. Moreover, based on a provided interpretation of a given PIT histogram, the user can modify its model to get better results.

## 5 DISCUSSION

The proposed methodological concept is designed to interpret PIT histograms automatically. They are decomposed into observation-generating and predictive distributions, allowing us to diagnose calibration problems of predictive models automatically. We employ a mixture density network, an interpreter, to perform this interpretation. We generate a synthetic data set of PIT histograms tailored to expected observation-generating and predictive distributions to train the interpreter. The interpreter approximates the observation-generating distribution that probably produced a given PIT histogram. This concept provides a powerful tool for diagnosing calibration problems of predictive models, offering the potential to enhance their calibration.

We showed that the concept works on real-world data sets, but still, we see several areas for improvement. Its limitation is that the synthetic data set of PIT histograms is based on a single predictive distribution. At the same time, in practice, we want to analyse models that output predictive various families of distributions. We would have to generate a specific synthetic data set for every family and train an interpreter with it. Although generating such synthetic data sets can be difficult, it is possible. Next, a simplification is that we have a single predictive distribution for all observations, while in practice, we would have a different predictive distribution based on input features for every observation. It looks like a small problem from our experiments, but we leave it for future research.

There has been much work on finding a method with the best predictive uncertainty, e.g. some are based on Bayesian theory (Hernandez-Lobato & Adams, 2015; Gal & Ghahramani, 2016), others on ensembles (Lakshminarayanan et al., 2017; Egele et al., 2022). We conclude our work by suggesting that it may be about something other than finding a method with the best predictive uncertainty. Our experiments suggest that choosing a correct predictive distribution might be more important than finding the best method. Our novel methodological concept is a step towards diagnosing calibration problems and thus identifying correct predictive distributions.

## REPRODUCIBILITY STATEMENT

The code underlying this work is available as supplementary material. The real-world regression data sets are publicly available in the UC Irvine Machine Learning Repository. The data preparation applied to these data sets is described in section 4.4. All values of hyperparameters are named individually in section 4.

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

## A PIT HISTOGRAM & CALIBRATION PLOT

We show that PIT histograms and calibration plots are equivalent, given $n$ predictive cumulative distribution distributions $F_i$ and corresponding observations $y_i$.

To construct a PIT histogram, we choose $k$ bin edges $0 = e_1 < \ldots < e_k = 1$. The PIT histogram displays bin counts $(b_1, b_2, \ldots, b_{k-1})$, where

$$b_j = |\{i | e_j \leq F_i(y_i) < e_{j+1}, i = 1, \ldots, n\}|.$$

The PIT histogram of a calibrated model is uniform.

To construct a calibration plot (Kuleshov et al., 2018), we choose $k$ confidence levels $0 \leq p_1 < \ldots < p_k \leq 1$. The calibration plot displays $\{(p_j, \hat{p}_j)\}_{j=1}^k$, where

$$\hat{p}_j = \frac{|\{i|F_i(y_i) < p_j, i = 1, \ldots, n\}|}{n}.$$

The calibration plot of a calibrated model has the diagonal line from $(0, 0)$ to $(1, 1)$.

It is evident that the two tools are equivalent. The main difference is that the calibration plot is cumulative, while the PIT histogram is not.

