# OpenReview forum: "Automatic Calibration Diagnosis: Interpreting Probability Integral Transform (PIT) Histograms"
_ICLR.cc/2024/Conference — Submitted to ICLR 2024_

### Official Review · Reviewer_ChB1 · 2023-10-31

**Soundness:** 1 poor
**Presentation:** 1 poor
**Contribution:** 1 poor
**Rating:** 1
**Confidence:** 5

**Summary:**

This paper considers the problem of evaluating the calibration level of a (probabilistic) regression model. The authors proposed to empirically obtain a distribution on the (conditional) CDF values on the observed target (i.e. PIT histogram) and use a mixture of the Gaussian model to approximate target distribution and compare the CDF value distribution with the empirical one as an indicator of miscalibration.

**Strengths:**

This paper made some efforts to explain its proposed idea of fitting the empirical CDF value histogram to check the calibration level.

**Weaknesses:**

This paper failed on multiple fronts to be considered a reasonable research paper.

The coverage of the field is minimal: The concept of calibration in regression tasks includes multiple established definitions (quantile calibration, threshold calibration, distribution calibration), while this paper only mentioned one related work on quantile calibration without giving a formal definition. Many well-known tools to measure miscalibration were ignored (calibration error, pin-ball loss, calibration test etc) or not correctly called out (reliability diagrams were only referred to as calibration plots without formal introduction).

The proposed method wasn't justified, and the setup was very limited: The proposed method was to fit a target distribution and then compare the CDF value distribution to check the calibration level, which seems to be redundant for multiple reasons. First, once the CDF value distribution / PIT histogram is obtained, it is already equal to the reliability diagram at hand, directly leading to metrics like calibration error. On the interpretability side, all the mixture model does is approximate the marginal distribution of the target value, which can be easily obtained via multiple approaches, including simple histograms. Furthermore, the paper only considered a constant predicted distribution on the target value, which could hardly be called a regression model in the first place.

**Questions:**

I wonder if the authors can suggest any reason to not to use reliability diagram and calibration error? they give the level of miscalibration and can be well-interpreted to see mismatch between the predicted percentiles and the marginal percentiles of the target distributions.

---

### Official Review · Reviewer_fJj1 · 2023-10-31

**Soundness:** 1 poor
**Presentation:** 1 poor
**Contribution:** 1 poor
**Rating:** 1
**Confidence:** 5

**Summary:**

The paper addresses the calibration problem associated with predictive distributions in machine learning models used for regression tasks. In particular, the authors assess calibration through the utilization of Probability Integral Transform (PIT) histograms and introduce an approach for their automated interpretation. They construct an interpreter by training a mixture density network on a synthetic dataset of PIT histograms. The methodology allows the identification of potential miscalibration issues by comparing the predictive distribution with the likely observation-generating distribution. The experiments are conducted on standard regression datasets, and the outcomes reveal improvement in the calibration of machine learning models.

**Strengths:**

Investigating PIT histograms to offer an easily understandable interpretation that aids in identifying calibration issues is a valuable and interesting direction of research.

**Weaknesses:**

- The contributions of this paper are quite limited. Section 3 is concise and lacks thorough (theoretical) justification for the proposed method.
- The choice of a neural network for the interpreter is unclear. The authors state that they want to represent the "inverse function of PIT" using the interpreter, which I understand as learning the inverse function of Quantile Recalibration [1]. Instead of using a neural network, this function could be computed analytically.
- The empirical evaluation is limited to three datasets, which is insufficient.
- The observed improvement in NLL and CRPS is attributable to the use of mixture predictions, which has been observed before in [2].
- The statement that "Optimising the sharpness of the distribution subject to its calibration is necessary." is too strong. Note that [3] only proposed this concept as a paradigm.
- It should be specified that one specific type of calibration, called probabilistic calibration in [3], is discussed.

- References
  - [1] Kuleshov, V. et al. Accurate Uncertainties for Deep Learning Using Calibrated Regression. ICML 2018.
  - [2] Dheur, V. et al. A Large-Scale Study of Probabilistic Calibration in Neural Network Regression. ICML 2023.
  - [3] Gneiting, T. et al. Probabilistic forecasts, calibration and sharpness. Journal of the Royal Statistical Society, 2017.

**Questions:**

See weaknesses.

---

### Official Review · Reviewer_2H7N · 2023-10-31

**Soundness:** 2 fair
**Presentation:** 3 good
**Contribution:** 2 fair
**Rating:** 3
**Confidence:** 4

**Summary:**

This paper proposes a PIM histogram and interpreter to measure the calibration error and find a way to data-generating regimes based on the PIT histogram. The use of PIM histogram is not surprising and standard approach in the calibration study, the use of interpreter is somewhat novel. Basically, if the calibration is achieved, the use of PIM is natural. The advanced contribution is that the author(s) proposed the interpreter learned based on the neural net and mixture of Gaussian. Simple experiments show the advantages of the proposed algorithms.

**Strengths:**

The experiments seem sufficient to validate the practical use of the PIT histogram, and how to interprete the non-calibrations. Most interesting part is to make the learning procedure the data-generating distribution through mixture of Gaussian among various methodologies. The use of GM is attractive and provides the insight for non-calibration issues.

**Weaknesses:**

The most weakness is too limited experiments. There is no case for the multi-modal density. Usually the density in the regression can be converged to unimodal in many cases. However, the authors emphasize the potential to address the multi-model problem. It it is not easy to find the proper real dataset, synthetic dataset can be considered. Also, large scale dataset is not examined. Usually, the calibration problem can be severe in large datasets, and the behaviors of calibration can differ w.r.t. size of datasets. The third is no applicability in learning process, which is required in recent studies.

**Questions:**

Q1: In data-generating regime, $y_{ij}$, $j$ is required. However, we cannot find these types of datasets in general exception of repeated measurements. When the data-generating regime is $y_i$ given $x_i$, is there no problem? How to assess the effect of $x_i$?

Q2: Can the number of elements in the mixture of Gaussian be tuned by any other strategy?

Q3: In figure 3, I cannot know how to calculate the negative log likelihood. Please explain how to calculate this value.

---

### Official Review · Reviewer_CsR9 · 2023-11-02

**Soundness:** 1 poor
**Presentation:** 4 excellent
**Contribution:** 2 fair
**Rating:** 3
**Confidence:** 3

**Summary:**

The submission presents a method to automatically diagnose calibration problems in conditional density estimation using probability integral transform histograms. This is based on creation of a synthetic dataset of training examples representing specific forms of miscalibration. Based on these training examples, which pair information on true densities and (mis-)estimated densities, a mixture density network is learned that given a histogram of the (mis-)estimated density, outputs an estimate of the true density. Based on examples, it is shown how this approach can be used to potentially diagnose miscalibration problems by considering three conditional density estimation models.

**Strengths:**

Calibration for distributional regression is an interesting problem and the paper's introduction to probability integral transform histograms is interesting.

**Weaknesses:**

It seems to me that this submission is in desperate need of some theory that can give some confidence in the capability of the proposed approach. Currently, there are only some examples for real-world datasets where estimated densities and (corrected) densities produced by the mixture density network are visually compared.

The synthetic data used to train the mixture density network is limited to using mixtures of two normals for the true densities and single normal densities as the (mis-)estimated densities. It seems highly unlikely that this data is sufficiently general to capture all the forms of miscalibration that can occur in practice, and it is unclear how one would produce such a dataset.

**Questions:**

Based on what is proposed mehtod, it seems that one should simply always use mixture density networks as the base estimators to avoid detection of miscalibration using the proposed method. Have you come across any scenarios where MDNs are miscalibrated, and could such a scenario be included in the paper?

---

### Meta-Review · Area_Chair_wBUo · 2023-11-27

**Metareview:**

All the reviewers were negative or very negative about the paper and they identified multiple areas for improvement.

**Justification For Why Not Higher Score:**

All the reviewers are negative about the paper.

**Justification For Why Not Lower Score:**

N/A

---

### Decision · Program_Chairs · 2024-01-16

Reject